# Efficient and stable visible-light-driven Z-scheme overall water splitting using an oxysulfide H$_2$ evolution photocatalyst

Lihua Lin[1], Yiwen Ma[1], Junie Jhon M. Vequizo [1], Mamiko Nakabayashi [2], Chen Gu[1], Xiaoping Tao[1], Hiroaki Yoshida [3,4], Yuriy Pihosh [5], Yuta Nishina [6], Akira Yamakata[7], Naoya Shibata [2], Takashi Hisatomi [1], Tsuyoshi Takata[1] & Kazunari Domen [1,5] ✉

So-called Z-scheme systems permit overall water splitting using narrow-bandgap photocatalysts. To boost the performance of such systems, it is necessary to enhance the intrinsic activities of the hydrogen evolution photocatalyst and oxygen evolution photocatalyst, promote electron transfer from the oxygen evolution photocatalyst to the hydrogen evolution photocatalyst, and suppress back reactions. The present work develop a high-performance oxysulfide photocatalyst, Sm$_2$Ti$_2$O$_5$S$_2$, as an hydrogen evolution photocatalyst for use in a Z-scheme overall water splitting system in combination with BiVO$_4$ as the oxygen evolution photocatalyst and reduced graphene oxide as the solid-state electron mediator. After surface modifications of the photocatalysts to promote charge separation and redox reactions, this system is able to split water into hydrogen and oxygen for more than 100 hours with a solar-to-hydrogen energy conversion efficiency of 0.22%. In contrast to many existing photocatalytic systems, the water splitting activity of the present system is only minimally reduced by increasing the background pressure to 90 kPa. These results suggest characteristics suitable for applications under practical operating conditions.

The utilization of renewable solar energy to produce clean, energetically-dense and storable hydrogen gas from water is expected to play an important role in the development of future green energy systems[1]. Among the many related technologies under development, photocatalytic water splitting using particulate semiconductors is regarded as a potentially viable means of generating renewable hydrogen on a large scale[2,3]. However, the solar-to-hydrogen energy conversion efficiency (STH) achievable with current technology is still lower than that required for practical applications[4,5]. The design of highly efficient photocatalyst materials and systems for water splitting is therefore an important aspect of achieving practical hydrogen production via photocatalytic water splitting.

Because visible light comprises more than 50% of the solar spectrum, much effort has been dedicated to the development of visible-light-responsive photocatalysts[6-8]. However, only few visible-light-responsive photocatalysts can realize one-step excitation overall water splitting, and most of them show low performance at present[9-11]. Therefore, efficient visible-light-driven photocatalysts have been

[1]Research Initiative for Supra-Materials, Interdisciplinary Cluster for Cutting Edge Research, Shinshu University, Nagano, Japan. [2]Institute for Engineering Innovation, The University of Tokyo, Bunkyo-ku, Tokyo, Japan. [3]Science and Innovation Center, Mitsubishi Chemical Corporation, Aoba-ku, Yokohama-shi, Kanagawa, Japan. [4]Japan Technological Research Association of Artificial Photosynthetic Chemical Process (ARPChem), Tokyo, Japan. [5]Office of University Professors, The University of Tokyo, Bunkyo-ku, Tokyo, Japan. [6]Graduate School of Natural Science and Technology, Okayama University, Kita-ku, Okayama, Japan. [7]Faculty of Natural Science and Technology, Okayama University, Kita-ku, Okayama, Japan. ✉e-mail: domen@chemsys.t.u-tokyo.ac.jp

employed primarily in conjunction with so-called two-step excitation Z-scheme overall water splitting (OWS) systems, due to the more stringent requirements associated with one-step excitation OWS[12,13]. Sulfides[14,15], (oxy)nitrides[16,17] and conjugated polymers[18–20] have been used as hydrogen evolution photocatalysts (HEPs), while oxides (both undoped and doped) have been applied as oxygen evolution photocatalysts (OEPs)[21–23], in Z-scheme OWS systems. Recently, oxysulfides have received increasing attention because these materials are able to absorb at longer wavelengths and exhibit superior stability compared with sulfide photocatalysts[11,24,25]. In particular, $Sm_2Ti_2O_5S_2$ (STOS), which can harvest sunlight up to 650 nm, has been studied as a photocatalyst since 2002[26,27]. Although one-step excitation OWS using STOS has not yet been reported, this material is applicable as the HEP in Z-scheme OWS systems[28,29].

Previous STOS-based Z-scheme OWS systems examined the use of $TiO_2$ or $WO_3$ as the OEP together with the $I_3^-/I^-$ redox couple as an electron mediator. Unfortunately, the STH was too low to be measured when $TiO_2$ was used as the OEP, while a value of only 0.003% was obtained with $WO_3$. This low performance was primarily attributed to the low activity of both the HEP and OEP in response to visible light and to the back reaction and competing reactions induced by the Pt cocatalyst and ionic redox couple, respectively. Therefore, drastic improvements in the efficiencies of STOS-based Z-scheme systems are required. This will necessitate increasing the intrinsic activity of the photocatalysts along with the use of surface modifications and electron mediators that do not promote undesired reactions.

The present work developed a highly active STOS catalyst exhibiting an apparent quantum yield (AQY) of 21.7% at 420 nm during the $H_2$ evolution reaction from aqueous methanol solutions. A Z-scheme system employing this high-performance STOS as the HEP, $BiVO_4$ (BVO) as a well-established OEP[30–34], and reduced graphene oxide (RGO) as a solid-state electron mediator[35,36] was found to split water in

a stable manner. This system was also more efficient than previous STOS-based systems, exhibiting an AQY of 7.0% at 420 nm and an STH of 0.22%, which was one of the best among the systems of $SrTiO_3$-$WO_3$, doped $SrTiO_3$-$BiVO_4$, TaON-$BiVO_4$ and $BaTaO_2N$-$WO_3$[37]. Notably, the performance was only minimally reduced even when the reaction was performed under gaseous Ar at near atmospheric pressure. These exceptional characteristics resulted from the enhanced activity of both the STOS and BVO, the exceptional electron transfer capability of the RGO, and the suppression of back reactions by surface modification with $Cr_2O_3$.

## Results and discussion
### Characterization of photocatalysts
The X-ray diffraction (XRD) pattern for the as-prepared STOS showed sharp diffraction peaks, indicating a high degree of crystallinity (Supplementary Fig. 1a)[38]. Images acquired using scanning electron microscopy (SEM) established that this material was composed of plate-like particles with sizes of 1-2 μm (Fig. 1a). Regular lattice fringes were clearly evident in high-resolution transmission electron microscopy (HR-TEM) images without any obvious evidence of dislocations or grain boundaries (Fig. 1b). These data, in combination with a selected area electron diffraction (SAED) analysis, confirmed the single crystalline nature of the STOS (Fig. 1c). The Sm, Ti, O and S present in the specimen were analyzed using X-ray photoelectron spectroscopy (XPS) and these same assessments indicated that residual flux components (Li, Ca, or Cl) were not present (Supplementary Fig. 2). Following surface modifications, $Ir^{4+}$, $Pt^0$, and $Cr^{3+}$ were all identified by XPS, indicating that $Cr_2O_3$, Pt and $IrO_2$ were deposited on the STOS surfaces (Supplementary Fig. 3), which was referred to as $Cr_2O_3$/Pt/$IrO_2$. Ir, Pt, and Cr species were also detected using high-angle annular dark-field scanning transmission electron microscopy (HAADF-STEM) and energy dispersive spectroscopy (EDS) mapping (Fig. 1d–f). The

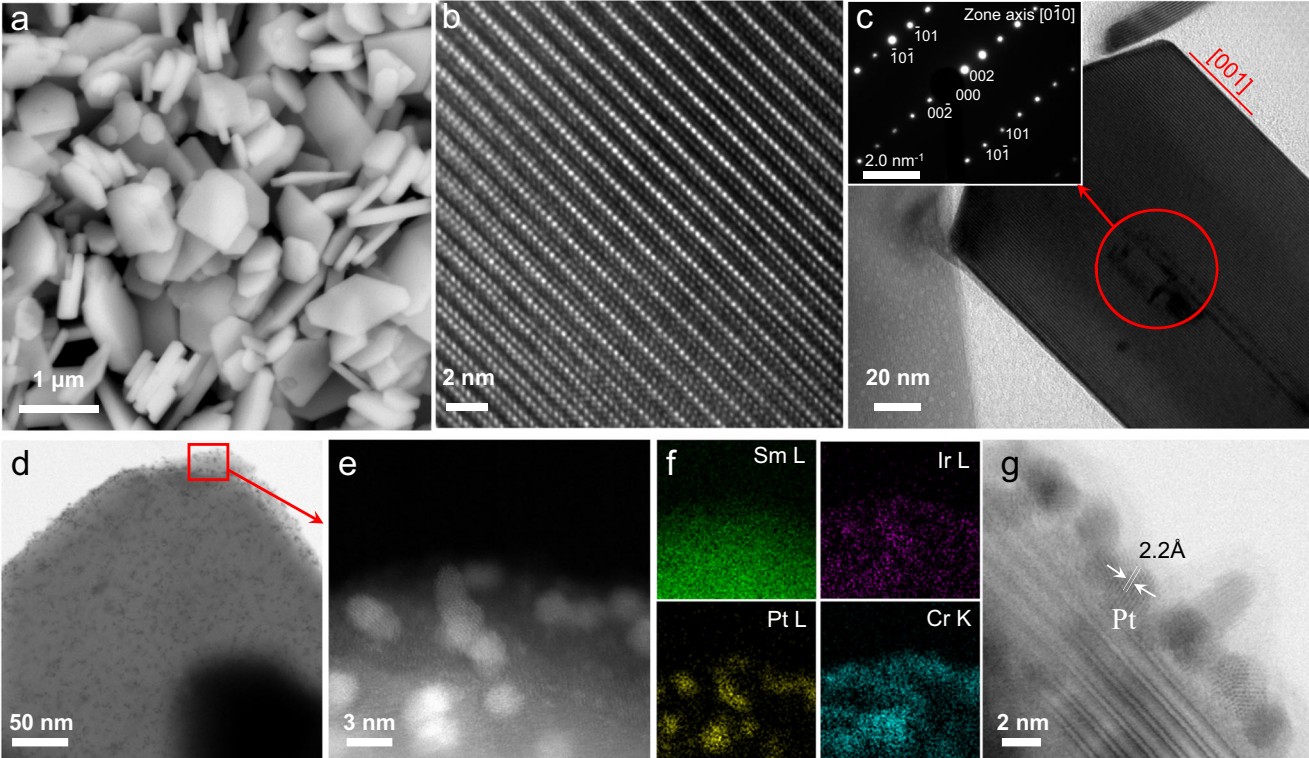

**Fig. 1 | Characterization of STOS. a** An SEM image of a STOS specimen. **b** Lattice fringes generated by the STOS as obtained using HR-TEM. **c** A TEM image of a STOS specimen and the corresponding SAED pattern. **d** A TEM image of a $Cr_2O_3$/Pt/$IrO_2$/ STOS specimen and **e** an enlarged image of the surface. **f**, EDS mapping of Sm, Ir, Pt, and Cr. **g** An HR-TEM image of the surface of a $Cr_2O_3$/Pt/$IrO_2$/STOS specimen.

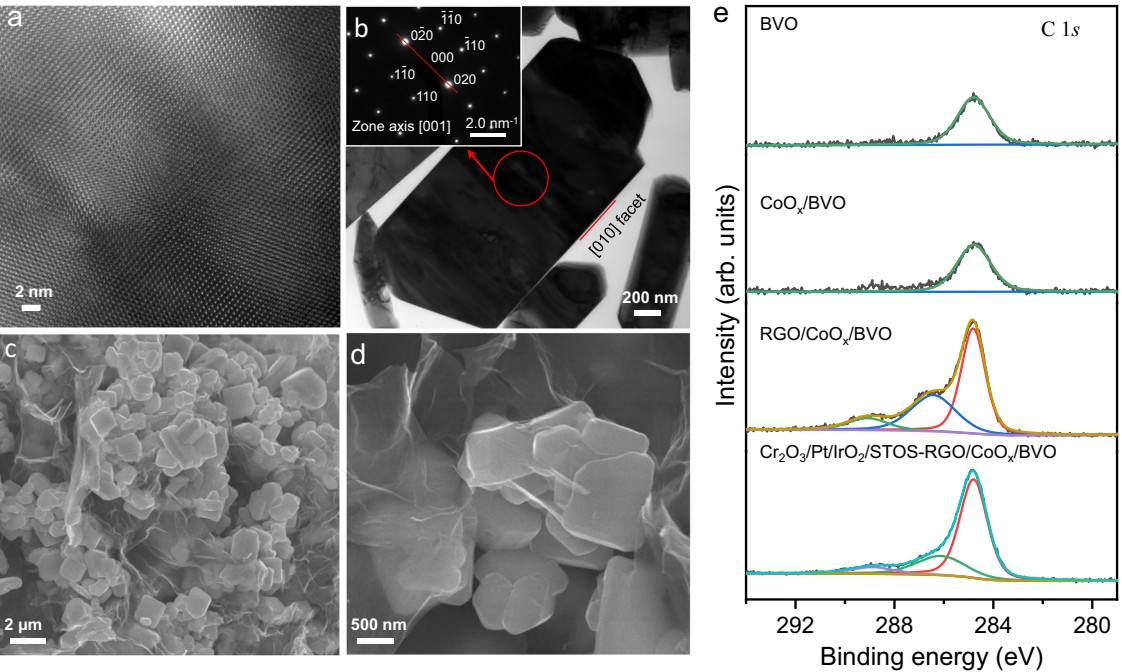

**Fig. 2 | Characterization of BVO. a** Lattice fringes obtained from BVO using HR-TEM. **b** A TEM image of a BVO specimen and the corresponding SAED pattern. **c, d** SEM images of an RGO/CoO$_x$/BVO specimen. **e** The C$1s$ XPS peaks of each sample.

IrO$_2$ particles sizes were determined to be less than 1 nm in size while the Pt particles were approximately 2-4 nm in size and appeared to be coated with Cr$_2$O$_3$ (Fig. 1g). In prior work, STOS prepared using a CsCl flux was found to contain Sm$_2$Ti$_2$O$_7$ and TiO$_2$ as impurities[39]. In addition, the sheet-like particles of STOS generated in previous research were thinner than those in the present study and exhibited irregular shapes, indicating a lower degree of crystallinity (Supplementary Fig. 4). The previous material also exhibited poor chemical stability, as evidenced by a strong odor of H$_2$S during the acid pre-treatment prior to surface modifications. The XPS analysis showed that the peaks of Sm $3d$ and Ti $2p$ were slightly shifted to higher and lower binding energy, respectively, compared with those in the STOS sample prepared with the CaCl$_2$/LiCl flux (Supplementary Fig. 5a, b). In addition, the peak intensity of adsorbed oxygen species was greatly increased (Supplementary Fig. 5c). Two additional S $2s$ peaks located at 231.4 and 227.5 eV were observed. The former can be ascribed to the formation of SO$_4^{2-}$, while the latter was assigned to the adsorbed H$_2$S as a strong odor of H$_2$S was noticed during the acid post-treatment (Supplementary Fig. 5d). Therefore, the surface state of STOS-CsCl was significantly different from STOS prepared with the CaCl$_2$/LiCl flux. Numerous dislocations were observed on (001) surface of STOS-CsCl and high resolution TEM analysis further confirmed the presence of defects inside the crystal (Supplementary Fig. 6). These results indicated that STOS prepared with the CsCl flux had poor crystallinity. Moreover, it was found that STOS could be obtained as the major product by using CaCl$_2$ or LiCl flux only. In the former case, impurity of TiO$_2$ was observed while the Sm$_2$Ti$_2$O$_7$ were found in the later one (Supplementary Fig. 7). In contrast to CsCl flux, no obvious difference was found by the XPS analysis (Supplementary Fig. 8), indicating that the surface state of STOS was unchanged regardless of the kinds of the flux reagents. However, the sheet-like STOS-CaCl$_2$ particles were larger and more irregular, forming agglomerates, compared with STOS prepared with the CaCl$_2$/LiCl mixture (Supplementary Fig. 9a). In addition, impurity of TiO$_2$ particles were also observed by the SEM. Similar results were found in the case of STOS-LiCl (Supplementary Fig. 9b). The higher melting points of CaCl$_2$ (772 °C) and LiCl (605 °C) than that

of the CaCl$_2$/LiCl eutectic mixture (475 °C) may account for the low crystallinity and photocatalytic activity.

An XRD pattern obtained from the BVO is presented in Supplementary Fig. 1b and is in agreement with data previously reported in the literature[31,34]. This material was confirmed to be highly crystalline by analyses using HR-TEM and SAED (Fig. 2a, b). After the photodeposition of CoO$_x$ on the BVO, a weak Co XPS signal was detected along with Bi, V and O peaks (Supplementary Fig. 10)[32]. Following photoreduction of GO on the CoO$_x$/BVO surfaces, wrinkled RGO sheets were observed by both SEM and TEM (Fig. 2c, d and Supplementary Fig. 11). The C $1s$ XPS spectrum acquired from the resulting RGO/CoO$_x$/BVO exhibited peaks centered at 284.8, 286.4 and 289.1 eV attributable to the C-C, C-O and C = O bonds of the RGO, respectively (Fig. 2e)[36].

**Photocatalytic performances**

Photocatalytic hydrogen and oxygen evolution half-reactions were carried out to assess the performance of the HEP and OEP. In these trials, almost no H$_2$ was evolved from an aqueous methanol solution using bare STOS, IrO$_2$/STOS or Cr$_2$O$_3$/IrO$_2$/STOS due to the lack of active sites. H$_2$ evolution was observed when employing the Pt/IrO$_2$-loaded STOS photocatalyst, and was dramatically enhanced by an additional modification with Cr$_2$O$_3$ (Supplementary Fig. 12a). The optimal loading amount of IrO$_2$, Pt, and Cr$_2$O$_3$ were found to be 0.5 wt %, 1.0 wt% and 0.5 wt%, respectively (Supplementary Fig. 12b–d). The H$_2$ evolution rate for Cr$_2$O$_3$/Pt/IrO$_2$/STOS was also found to increase with increasing mass of photocatalyst used, reaching a maximum of 2.7 mmol·h$^{-1}$ H$_2$ during the first hour of the process in the case that 0.2 g of the photocatalyst was used in conjunction with the present reaction conditions (Fig. 3a). Further increasing the concentration of the photocatalyst powder will block the incident light due to the scattering at the top part of the suspension, which would lower the number of photons absorbed by the photocatalyst and therefore decrease H$_2$ evolution performance when the amount of photocatalyst was more than 0.2 g. A plot of AQY as a function of the irradiation wavelength closely matched the light absorption profile for STOS (Fig. 3b). The AQY value at 420 nm was 21.7% and so was 36 times that

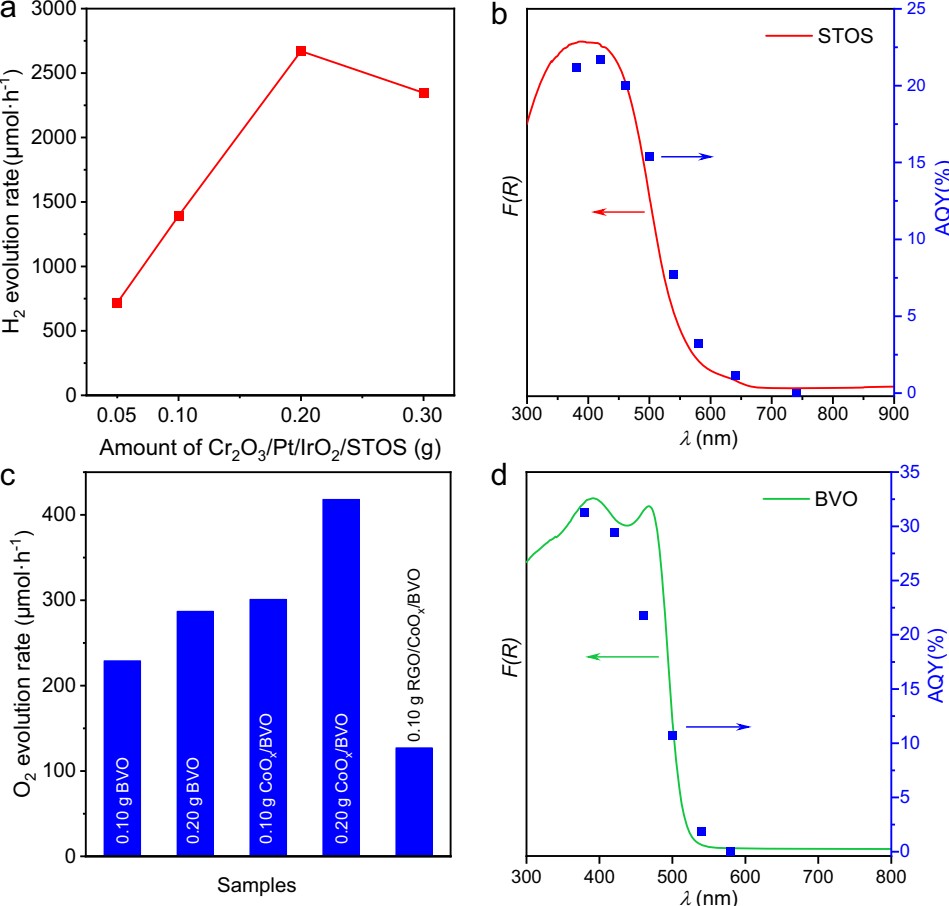

**Fig. 3 | Photocatalytic H$_2$ and O$_2$ evolution activities. a**, H$_2$ evolution rates over Cr$_2$O$_3$/Pt/IrO$_2$/STOS in aqueous methanol as a function of catalyst mass and **b** corresponding AQY values for Cr$_2$O$_3$/Pt/IrO$_2$/STOS (0.2 g) at different wavelengths together with DRS data. **c** O$_2$ evolution rates over BVO, CoO$_x$/BVO, and RGO/ CoO$_x$/BVO in aqueous AgNO$_3$ solutions and **d** corresponding AQY values for CoO$_x$/BVO (0.2 g) at different wavelengths together with DRS data.

previously reported for STOS loaded with Pt and IrO$_2$. Cr$_2$O$_3$/Pt/STOS also showed considerable H$_2$ evolution ability but was not as efficient as Cr$_2$O$_3$/Pt/IrO$_2$/STOS in this regard, indicating that the co-loading of IrO$_2$ with Pt promoted H$_2$ generation. It should also be noted that no H$_2$ evolution activity was obtained when using STOS prepared with a CsCl flux after the same surface modifications, as a consequence of the lower crystallinity and poor stability of this material. Steady H$_2$ production was observed under the same reaction conditions in the case of STOS·CaCl$_2$ and STOS·LiCl with similar performance, but the H$_2$ evolution rates were much lower compared with STOS prepared by CaCl$_2$/LiCl eutectic mixture due to the poor crystal quality (Supplementary Fig. 12e).

In OEP trials, the initial O$_2$ evolution rate was 287 µmol·h$^{-1}$ with a 0.2 g quantity of bare BVO in the presence of AgNO$_3$ as a sacrificial agent. Following the photodeposition of CoO$_x$, this rate increased to 418 µmol·h$^{-1}$ (Fig. 3c). The AQY values for the CoO$_x$/BVO combination at 420, 460 and 500 nm were determined to be 29.5%, 21.8% and 10.7%, respectively (Fig. 3d). Notably, photodeposition of RGO on CoO$_x$/BVO lowered the O$_2$ evolution rate even though this modification was essential to the construction of an effective Z-scheme OWS system.

To ensure the formation of Z-scheme water splitting system, the conduction band minimum (CBM) of STOS was determined by Mott-Schottky plot and it was located at around −0.6 V vs. NHE (Supplementary Fig. 13a). According to the bandgap of 1.9 eV obtained by UV-visible diffuse reflectance spectroscopy (DRS), the valance band maximum (VBM) located at around 1.3 V vs. NHE. In combination with the reported band position of well-developed BVO[32,33], it was found

that the band alignment allows the charge transfer in the Z-scheme manner (Supplementary Fig. 13b). Consequently, a Z-scheme OWS system was produced based on the HEP and OEP prepared as described above. In a typical reaction, 0.05 g of Cr$_2$O$_3$/Pt/IrO$_2$/STOS and 0.1 g of RGO/CoO$_x$/BVO were suspended in 150 mL of pure water with continuous stirring. The activity of this system was found to improve over time during the first 12 h (Supplementary Fig. 14). This result suggests that the HEP and RGO-loaded OEP underwent photocatalytic reaction to form chemical bonds similar to those connecting RGO and BVO. SEM images demonstrated that RGO gradually covered the Cr$_2$O$_3$/Pt/IrO$_2$/STOS and CoO$_x$/BVO particles during this induction period, establishing a connection between the HEP and OEP (Fig. 4a and Supplementary Fig. 15). From the images of EDX mapping, the STOS and BVO can be identified by the distribution of Sm and Bi elements, respectively (Supplementary Fig. 16). Besides, in order to clearly observe RGO sheets, a secondary electron detector was used for imaging. When a back-scattering electron detector was used for imaging, it is possible to distinguish particles of STOS and BVO by the grayscale contrast (Supplementary Fig. 17). This is because the signal for back-scattered electrons becomes stronger with the increased atomic number of the element in the compound. The atomic number of Bi is greater than Sm and therefore BVO particles appear brighter than STOS particles. After the induction period, the reaction system was evacuated and Ar was introduced again to a specific pressure. The maximum H$_2$ and O$_2$ evolution rates in the initial stage of this process under visible light were 240 and 115 µmol h$^{-1}$, respectively. An AQY value of 7.0% was achieved in response to 420 nm monochromatic

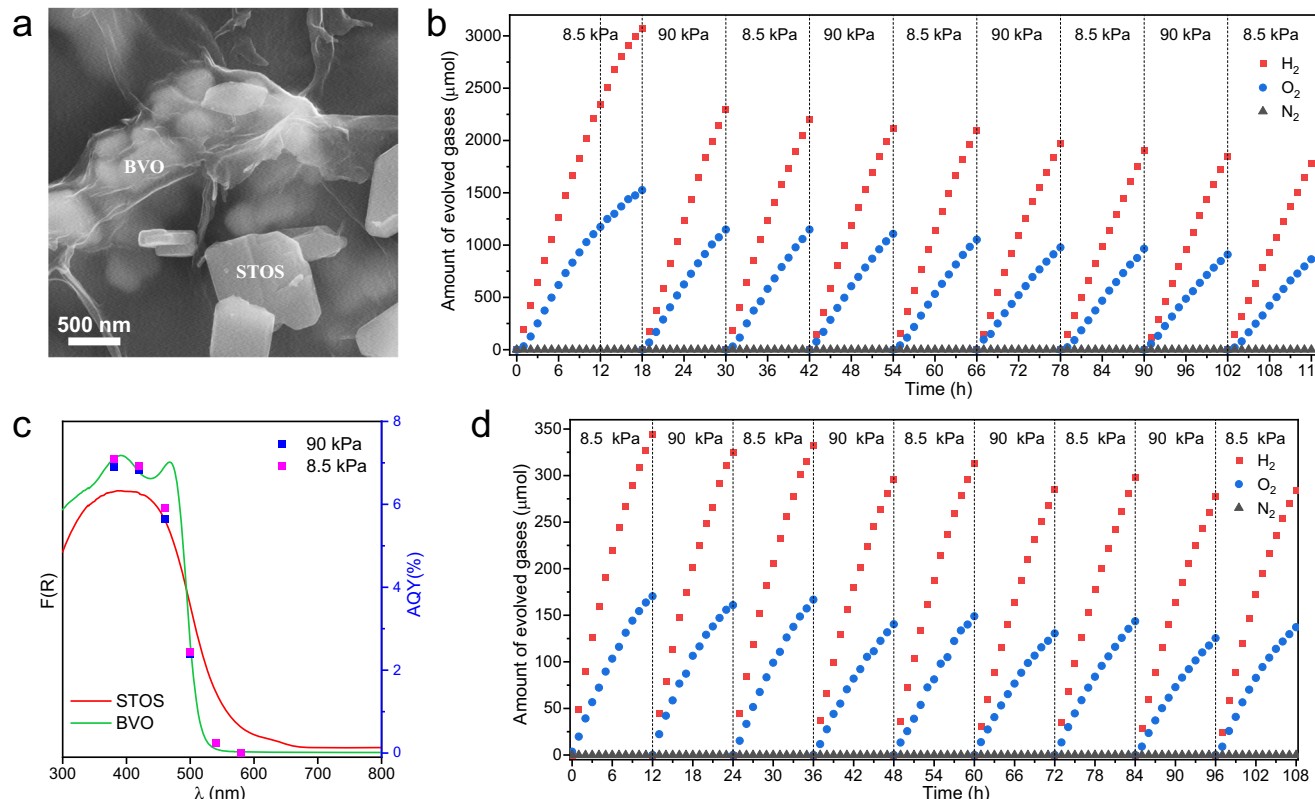

**Fig. 4 | Photocatalytic OWS reaction of a Z-scheme system. a** SEM image of the Cr$_2$O$_3$/Pt/IrO$_2$/STOS-RGO/CoO$_x$/BVO after the induction period. **b** The evolution of gases from the Z-scheme system over time at different Ar background pressures in response to >420 nm light. **c** AQY values obtained from the Z-scheme system under

8.5 and 90 kPa Ar at different wavelengths together with the DRS data for the STOS and BVO. **d** The evolution of gases from the Z-scheme system over time at different Ar background pressures in response to irradiation with an AM1.5 G solar simulator.

light (Fig. 4c), which was 7 times higher than that for a system comprising Ru/SrTiO$_3$:Rh and RGO/BiVO$_4$[36]. In addition, the activity was enhanced by increasing the GO loading from 0.3 to 0.8 wt% with respect to the mass of BVO. Further increasing the amount of GO did not provide any additional performance gains (Supplementary Fig. 18a). According to the elemental analysis, the loading amount of carbon was gradually increased with the increased amount of added GO (Supplementary Table 1). Not all the GO was photodeposited at higher added amount and the excess GO was simply removed during the centrifugation procedure. In the absence of GO on the CoO$_x$/BVO surface, the activity of the Z-scheme system became two orders of magnitude lower because the transfer of photogenerated carriers between Cr$_2$O$_3$/Pt/IrO$_2$/STOS and CoO$_x$/BVO particles became very inefficient. Notably, the activity of the present Z-scheme OWS system was highly dependent on the type of GO that was employed, varying from 7 to 240 μmol·h$^{-1}$ (based on the H$_2$ evolution rate) as the material was changed, as shown in Supplementary Fig. 18b. According to an SEM analysis, the activity was maximized in the case that the RGO was present as large sheets, which presumably favored intimate contact with the OEP and HEP (Supplementary Fig. 18c–e).

### Transient absorption spectroscopy

The Z-scheme system was also investigated using transient absorption spectroscopy (TAS) to monitor charge transfer from the photocatalysts to the co-catalysts and from the OEP to the HEP[40]. The TAS signals obtained from the bare STOS at different delay times exhibited broad absorption over the entire wavelength range (Fig. 5a and Supplementary Fig. 19). The photocarriers in this material survived for longer than 4 ms, indicating that charge carriers would likely be able to migrate to the surface of the catalyst (Fig. 5b). IrO$_2$-loaded STOS was also examined and generated similar spectra to those of the bare

sample, implying that photocarriers were present in the excited STOS. The time-dependent decay profile for electrons monitored at 5000 nm showed increased signal intensity after loading IrO$_2$. Hence, photoinduced holes in the STOS were likely captured by the IrO$_2$, leading to an increased electron population in the material[41–43]. In contrast, the TAS intensity was quenched significantly following the sequential deposition of Pt and Cr$_2$O$_3$, suggesting a reduced electron population in STOS and therefore enhanced electron capture by co-catalyst. These results are in agreement with the significant improvement in photocatalytic H$_2$ generation exhibited by STOS after surface modifications (Supplementary Fig. 12a). In the analysis of the BVO, photoinduced surface trapped holes were monitored at 19,800 cm$^{-1}$ (505 nm) based on previously reported studies of this compound[44–46]. Loading of CoO$_x$ as an OER co-catalyst on the BVO surface was found to reduce the TAS signal intensity at 505 nm and accelerate the decay of photoexcited holes in the BVO, indicating enhanced hole capture by the co-catalyst. The signal decay was further accelerated after loading the catalyst with RGO, implying that holes and electrons were captured by the RGO such that recombination was enhanced (Supplementary Fig. 20). This result was in agreement with the decreased O$_2$ evolution activity observed following photodeposition of RGO on CoO$_x$/BVO.

In trials assessing the Z-scheme system, a 470 nm pump was used to photoexcite both the HEP and OEP. The TAS signal observed at 5000 nm on the microsecond-millisecond time scale was attributed to electrons in STOS because a much weaker signal intensity was obtained when BVO alone was irradiated (Supplementary Fig. 21). The decay of photoexcited electrons in the Z-scheme system was prolonged compared with that in Cr$_2$O$_3$/Pt/IrO$_2$/STOS (Fig. 5c). This difference is attributed to the Z-scheme recombination of electrons from BVO with holes from STOS, meaning that the recombination of electrons and holes in STOS was suppressed. This effect allowed for the

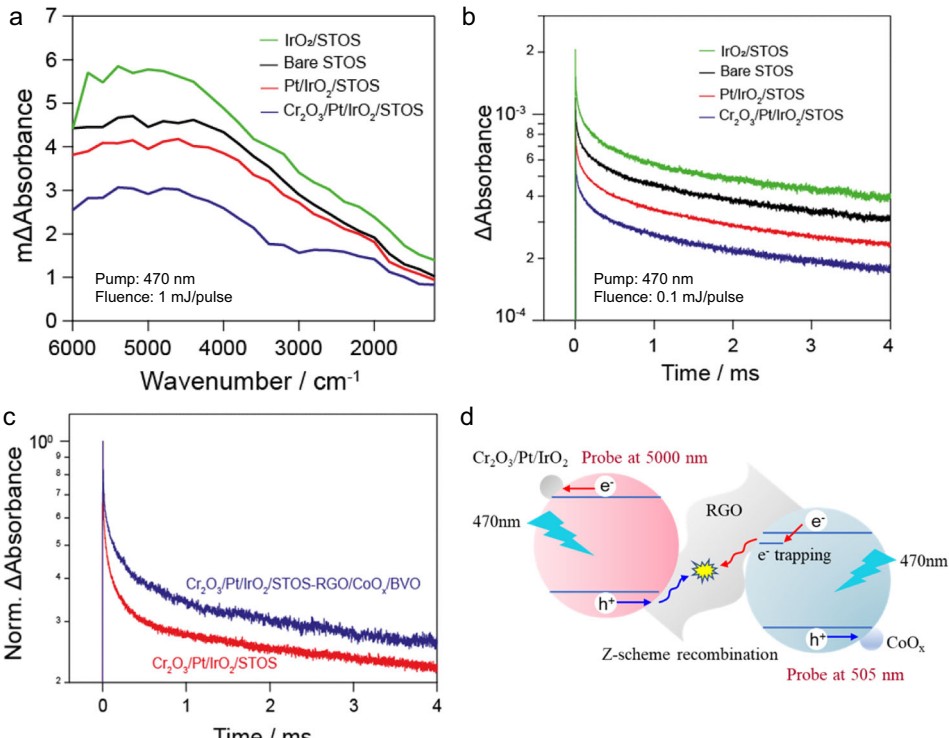

**Fig. 5 | TAS data. a** Mid-IR transient absorption spectra and **b** decay kinetics at 5000 nm obtained from the bare STOS and STOS loaded with $IrO_2$, $Pt/IrO_2$ and $Cr_2O_3/Pt/IrO_2$. **c** Decay kinetics of carriers in the $Cr_2O_3/Pt/IrO_2/STOS$ and $Cr_2O_3/Pt/$ $IrO_2/STOS-RGO/CoO_x/BVO$. **d** A diagram summarizing the charge transfer mechanism between the HEP and OEP in a Z-scheme system.

efficient capture of electrons and holes by the hydrogen and oxygen evolution co-catalysts, respectively, as illustrated in the scheme presented in Fig. 5d. Following selective photoexcitation of the HEP using a 560 nm pump, photoexcited electrons decayed faster in the Z-scheme system compared with those in $Cr_2O_3/Pt/IrO_2/STOS$ (Supplementary Fig. 22a) as a result of the absence of photogenerated electrons from BVO capable of recombining with holes in STOS. Consequently, the recombination of electrons and holes photogenerated in the bulk STOS or on the RGO was accelerated (Supplementary Fig. 22b). These findings establish a Z-scheme water-splitting mechanism that proceeds based on photoexcitation of both the HEP and OEP. To support the enhancement of charge transfer by RGO in the Z-scheme system, the TAS was measured after steady-state irradiation in the absence and presence of water vapor in a nitrogen atmosphere for the Z-scheme system with and without RGO (Supplementary Fig. 23). Note that the accumulated electrons can participate in the water reduction process in the presence of water vapor. In the absence of RGO, the decay of electrons was hardly accelerated by the introduction of water vapor, indicating a low efficiency to the water reduction reaction by photoexcited electrons. Interestingly, when RGO was incorporated into the Z-scheme system, the decay of electrons was much accelerated by the exposure to water vapor. This means that the reactivity of the accumulated electrons for water reduction was significantly enhanced by the RGO promoting the formation of the Z-scheme. These results demonstrated the connection of HEP and OEP with RGO can significantly promote the charge transfer between HEP and OEP.

## Factors enhancing photocatalytic activity

Control experiments were carried out in order to further investigate the mechanism responsible for the enhanced activity of the present catalyst system. In these trials, the solid-state electron mediator RGO was replaced by either $I_3^-/I^-$ or $Fe^{3+}/Fe^{2+}$ acting as an ionic couple, both of which had been commonly used in previous work[29,47,48]. Using $I_3^-/I^-$,

only $H_2$ evolution was observed throughout a 12 h photoreaction, indicating that the water oxidization reaction did not proceed in this system (Supplementary Fig. 24a). Although the OWS reaction proceeded with the $Fe^{3+}/Fe^{2+}$ ionic couple, the activity was nearly half that obtained using RGO as the electron mediator. In addition, the gas evolution rate was found to decrease as the photoreaction time was prolonged (Supplementary Fig. 24b). This phenomenon is attributed to the low pH of 2.3 used in conjunction with the $Fe^{3+}/Fe^{2+}$ ionic couple, which was detrimental to the stability of the co-catalysts and $Cr_2O_3$. These results indicate that RGO was a more efficient and stable means of promoting the transfer of electrons from the OEP to the HEP. In additional trials, STOS loaded with $Pt/IrO_2$ or $Cr_2O_3/Pt$ was used as the HEP to assess the effect of the surface modification methods on performance. The Z-scheme OWS reaction did not proceed when $Pt/IrO_2/$ STOS was used as the HEP, which can be ascribed to the increased progress of the back reaction in the absence of $Cr_2O_3$[49]. Significant $H_2$ and $O_2$ evolution rates with the expected stoichiometric molar ratio of 2:1 were obtained in experiments using $Cr_2O_3/Pt/STOS$ as the HEP, although the activity was lower than that observed with the $Cr_2O_3/Pt/$ $IrO_2/STOS$ (Supplementary Fig. 25a). It is evident that Pt and $Cr_2O_3$ were required to create $H_2$ evolution sites and inhibit the back reaction, respectively, while the role of $IrO_2$ was explored by the $O_2$ evolution of STOS with different surface modifications. Significantly improved activity of $O_2$ evolution was observed when the $IrO_2$ was loaded (Supplementary Fig. 25b). The enhancement of activity in both the $O_2$ and $H_2$ evolution half-reactions indicates that $IrO_2$ can capture photogenerated holes and therefore promote charge separation, resulting in the enhanced performance of HEP in overall water splitting. However, in $H_2$ evolution half reaction, $IrO_2$ was not an efficient co-catalyst for oxidizing the sacrificial reagent. Therefore, the $H_2$ evolution rate was not significantly enhanced after coloading of $IrO_2$. Overall, the trends observed in the Z-scheme OWS reaction activity when using HEP specimens having different surface modifications were the same as those seen in the corresponding $H_2$ evolution half-

reaction rates. Therefore, the high activity of the HEP and the functioning of the RGO as an efficient solid-state electron mediator together with suppression of the back reaction were the key factors accounting for the higher OWS performance.

The effect of the background pressure on the water-splitting activity was additionally investigated. Interestingly, the present Z-scheme system showed similar water-splitting activities and AQY values at Ar background pressures of 8.5 and 90 kPa (Fig. 4b, c). Therefore, the background pressure had little effect on this system, which is favorable for the application of an as-constructed Z-scheme system under practical operating conditions. The activity was seen to gradually decrease over time (by 24% after 114 h) regardless of the pressure, although $H_2$ and $O_2$ were continually evolved at the expected stoichiometric molar ratio of 2:1. After a 114 h photoreaction under visible light, the sample was captured by filtration and dried in preparation for characterization. XRD patterns were acquired from both the STOS and BVO phases (Supplementary Fig. 26). The XPS peaks generated by the constituent elements were found to be similar before and after the reaction, confirming the stability of the photocatalysts and co-catalysts (Supplementary Fig. 27). However, the adhesion of the photocatalyst powder to the reactor wall was visually confirmed after photoreaction for one run (12 h) (Supplementary Fig. 28). The photocatalyst powder in the solution was carefully collected by filtration, followed by drying in the vacuum oven. The procedure was repeated three times and the average amount of remaining photocatalyst was 146.8 mg. This means that in one run of the photoreaction for 12 h, 3.2 mg of the photocatalyst would adhere to the reactor wall. After each run, a similar amount of the photocatalyst powder would adhere to the reactor wall. During 8 runs for a total of 114 h, the amount of powder lost would be around 24 mg. Therefore, the evident decrease in activity was primarily attributed to the adhesion of the photocatalyst powder to the reactor wall rather than to (photo)chemical degradation of the components of the Z-scheme system. The STH value for the Z-scheme OWS system was determined to be 0.22% under simulated sunlight (AM 1.5 G) and so was 73 times higher than that reported for a prior Z-scheme system consisting of STOS synthesized using a flux as the HEP and $I_3^-/I^-$ as the electron mediator couple[29]. Again, the background pressure had little effect on performance and an STH of 0.21% was achieved under Ar at a background pressure of 90 kPa (Fig. 4d).

## Fabrication of Z-scheme OWS sheets
Although the suspension system concept is the simplest approach to promoting OWS on the lab-scale, water-splitting panels incorporating fixed photocatalysts are better suited to the construction of large-scale reactors[4,50,51]. Therefore, in the present work, a Z-scheme system was fixed on a substrate to fabricate particulate photocatalyst sheets for the OWS reaction. It is notable that these photocatalyst sheets exhibited comparable activity to the suspension system under similar reaction conditions (Supplementary Fig. 29a). Furthermore, these sheets maintained 89% of their original activity under a 90 kPa background pressure compared with that under a background pressure of 8.5 kPa, and also retained 73% of their original performance after a 108 h photoreaction (Supplementary Fig. 29b). Although the activity was lower than the best performance shown by the suspension system, these results at least support the feasibility of scaling-up the present Z-scheme system.

## Performance-limiting factors
At present, the STH value for the Z-scheme system is much lower than that required for practical $H_2$ production by photocatalytic water splitting, and so dramatic performance improvements are needed[5]. The variations in the AQY data for the OWS reaction (Fig. 4c) closely match the light absorption characteristics of BVO rather than those of STOS. Therefore, it is considered that the utilization of sunlight in the current Z-scheme water-splitting

system is limited by the OEP, which has a relatively wide band gap[52]. The incorporation of an OEP capable of absorbing longer wavelength light could be an effective approach to improving the performance. In addition, the $H_2$ evolution rate during the Z-scheme OWS reaction was barely affected by increasing the background pressure based on introducing $O_2$ while the $O_2$ evolution rate was significantly decreased upon adding $H_2$ (Supplementary Figs. 30a, b). These results indicate the occurrence of the hydrogen oxidation reaction ($H_2 \rightarrow 2H^+ + 2e^-$) with increases in the $H_2$ partial pressure within the reaction environment, leading to charge recombination through cycles of water reduction and hydrogen oxidation under illumination (Supplementary Fig. 30c)[53]. In fact, the occurrence of hydrogen oxidation was also noted in conjunction with the hydrogen evolution reaction from aqueous methanol solutions. Assuming a reverse reaction with a first-order dependence on the hydrogen partial pressure, it was possible to fit a plot of hydrogen evolution over time (Supplementary Fig. 30d). An increase in the amount of $Cr_2O_3/Pt/IrO_2/STOS$ also increased the amount of Pt co-catalyst that accelerated the hydrogen oxidation reaction, which competes with the $H_2$ evolution forward reaction. This was another reason accounting for the decreased $H_2$ evolution rate when the amount of photocatalyst exceeded 0.2 g (Fig. 3a). Moreover, the amounts of gaseous $H_2$ and $O_2$ were found to decrease gradually in the absence of light, providing further evidence for the occurrence of the back reaction in the Z-scheme system (Supplementary Fig. 31a). The back reaction evidently occurred more rapidly on the $Pt/IrO_2/STOS$ in the absence of a $Cr_2O_3$ coating but did not take place in trials using the $IrO_2/STOS$, $Cr_2O_3/IrO_2/STOS$ or $RGO/CoO_x/BVO$ systems (Supplementary Figs. 31b-e). Clearly, the Pt component of the co-catalyst promoted the backward reaction. Although photodeposition of $Cr_2O_3$ on the Pt slowed the rates of $H_2$ and $O_2$ consumption, the backward reaction still occurred because the $H_2$ and $O_2$ were able to penetrate the imperfect $Cr_2O_3$ shells to make contact with the Pt co-catalyst (Supplementary Figs. 31f and g). The reverse reaction was further inhibited by increasing the amount of photodeposited $Cr_2O_3$ beyond 0.5 wt% (Supplementary Fig. 32a). However, at the same time, the $H_2$ evolution performance was gradually decreased with increasing the $Cr_2O_3$ loading amount (Supplementary Fig. 32b). Concurrently, the performance of overall water splitting was also decreased (Supplementary Fig. 32c). Therefore, photodeposition of 0.5 wt% Cr was an optimal amount to balance forward and reverse reaction. Additionally, the Z-scheme OWS performance was decreased with increasing amounts of HEP and OEP in the Z-scheme system (Supplementary Fig. 33). This result was in contrast to the behavior of the corresponding half reactions, for which the $H_2$ and $O_2$ evolution rates increased with increasing amounts of photocatalysts within specific certain ranges (Fig. 3a, c). These phenomena can be attributed to the enhanced back reaction occurring in the Z-scheme OWS with an increased amount of HEP. Accordingly, further suppression of the backward reaction on the HEP would be another means of enhancing the overall performance of the Z-scheme system. In the case of the sheet system, the proportion of the photocatalyst particles shaded by other particles was increased in the particulate layer. This shaded photocatalyst fraction did not promote the Z-scheme OWS reaction but did promote the backward reaction (induced by the Pt component in the co-catalyst) and so reduce the overall activity. Therefore, optimization of the sheet fabrication process is an essential aspect of improving performance.

In summary, a high-performance STOS specimen characterized by a high degree of crystallinity and long carrier lifetimes was prepared by a flux-assisted method. An AQY of 21.7% at 420 nm was achieved with regard to the $H_2$ evolution half-reaction after surface modification with

$IrO_2$, Pt, and $Cr_2O_3$. A Z-scheme system based on this material as the HEP, BVO as the OEP, and RGO as the solid-state electron mediator exhibited an STH of 0.22% during the OWS reaction. This value was 73 times higher than that obtained previously from a Z-scheme system constructed with STOS and $WO_3$, owing to the enhanced activity of the present HEP and OEP and the efficient electron transfer from the OEP to the HEP via the RGO. The water-splitting activity of this system was almost independent of the Ar gas pressure because backward reactions were largely suppressed by the surface modification with $Cr_2O_3$. A prototype panel-type OWS system suitable for large-scale applications was fabricated based on this Z-scheme system and showed comparable activity to that of the suspension system. This work demonstrates that enhancing the intrinsic activity of the HEP can significantly promote the overall performance of a Z-scheme OWS system. The importance of charge transfer between HEP and OEP was confirmed by TAS analysis and the back reaction was studied in detail, which will guide the future development of Z-scheme system. This study also indicates that, based on additionally improving the activity of the OEP and inhibiting the back reaction together with the reduced cost of photoreactor and co-catalysts, a high-performance Z-scheme OWS system that fulfills the requirements for large-scale applications could be achieved.

## Methods

### Preparation of STOS powder by flux method

STOS powder was synthesized by combining the precursors $Sm_2O_3$ (0.489 g, Wako), $Sm_2S_3$ (0.215 g, HIGH PURITY CHEMICALS), and $TiO_2$ (0.295 g, RARE METALLIC Co., Ltd.) at a 1:2:6 molar ratio followed by grinding in a glove box filled with $N_2$. Subsequently, 10.0 wt% sulfur powder (HIGH PURITY CHEMICALS) was added to the precursors to produce a sulfur-rich environment. Following this, a $LiCl/CaCl_2$ flux (3.0 g, 42.4:57.6 mass ratio, Wako) was added to the mixture at a 3:1 mass ratio relative to the original mixture mass, and the entire mixture was again subjected to grinding. The as-prepared powder was then sealed in a quartz tube under vacuum and calcined at 973 K for 24 h. After being allowed to cool naturally to ambient temperature, the product was dispersed in deionized (DI) water to dissolve the $LiCl/CaCl_2$ flux, and the resulting STOS powder was obtained by filtration and then dried in a vacuum oven. Excess sulfur was removed from the product by calcination of the STOS in the air at 473 K for 1 h followed by washing in 47% $H_2SO_4$ (45 mL, Wako) for 15 min. Finally, the powdered product was washed several times with DI water, removed by filtration and dried in a vacuum oven. The preparation procedure of STOS with different flux was the same as the $LiCl/CaCl_2$ flux but replaced the eutectic mixture with CsCl (3.0 g, Wako) or $CaCl_2$ (3.0 g, Wako) or LiCl (3.0 g, Wako), in which the obtained samples were named as STOS-CsCl, STOS-$CaCl_2$ and STOS-LiCl, respectively. Unless noted otherwise, the term "STOS" refers to the sample prepared by $LiCl/CaCl_2$ flux.

### Surface modification of STOS

A 0.2 g quantity of the STOS powder was dispersed in 15 mL $H_2O$ followed by the addition of 0.5 mL aqueous $IrCl_3 \cdot 3H_2O$ solution (3.669 mg/mL, TOKYO CHEMICAL INDUSTRY Co., Ltd.) sufficient to give an Ir concentration of 0.5 wt% with respect to the amount of STOS. The resulting mixture was transferred to a glass vial and then heated in a microwave reactor at 423 K for 15 min. The $IrO_2$/STOS mixture was then dispersed in 15 mL ethylene glycol (Wako) and an aqueous solution of $H_2PtCl_6 \cdot 6H_2O$ (Kanto) was added sufficient to give a 1.0 wt% Pt concentration with respect to the amount of STOS. The solution was transferred to a glass vial and heated in a microwave reactor at 423 K for 15 min. Immediately following, 0.5 wt% $Cr_2O_3$ was photodeposited on the Pt/$IrO_2$/STOS using $K_2CrO_4$ (Wako) as the precursor in 150 mL of a 10.0 vol% aqueous methanol solution. This step was performed with full-spectrum irradiation by a 300 W Xe lamp, similar to the conditions

applied during the photocatalytic $H_2$ evolution trials described below. After a 1 h photoreaction, the $Cr_2O_3$/Pt/$IrO_2$/STOS was collected by filtration and washed several times with DI water.

### Preparation of BVO powder by hydrothermal method

A precursor solution was prepared by dissolving 10.0 mmol of $NH_4VO_3$ (1.170 g, Wako) and 10.0 mmol of $Bi(NO_3)_3 \cdot 5H_2O$ (4.851 g, Wako) in a 2.0 M nitric acid solution (60 mL, Wako), the pH of which was adjusted to approximately 0.5 by adding an ammonia solution (28.0-30.0 wt%, Wako). The resulting solution was stirred for 2 h, during which time a pale yellow precipitate formed. This product was transferred to a Teflon-lined stainless steel autoclave for hydrothermal treatment at 473 K for 24 h. After natural cooling, the BVO powder was obtained by filtration and drying in a vacuum oven.

### Loading cobalt co-catalyst onto BVO

A cobalt co-catalyst was loaded onto the BVO using a photodeposition method. In this process, 0.2 g of BVO was dispersed in 150 mL of a 50 mM phosphate buffer solution (pH 6.0) containing $Co(NO_3)_2 \cdot H_2O$ (Kanto) such that the final concentration of Co was 0.5 wt% with respect to the amount of BVO. The suspension was then irradiated with full-spectrum light generated by a 300 W Xe lamp for 1 h, similar to the photocatalytic $H_2$ evolution procedure described below. After removal by filtration and washing, $CoO_x$/BVO was obtained.

### Photodeposition of RGO

Four types of lab-made (LM) GO were used during photodeposition trials. These are referred to as LM-GO-I, LM-GO-II, LM-GO-III, and LM-GO-IV herein. LM-GO-I was prepared by adding graphite (SP-1, Bay Carbon Inc.; 3.0 g) to $H_2SO_4$ (75 mL) followed by slowly adding $KMnO_4$ to give a graphite:$KMnO_4$ mass ratio of 1:3 with stirring at 200 rpm and cooling to 10 °C. The mixture was subsequently held at 35 °C for 2 h before quenching by the addition of $H_2O$ (75 mL) with vigorous stirring and cooling to ensure that the temperature did not exceed 50 °C. Following this, an $H_2O_2$ solution (30%, 7.5 mL) was slowly added with continuous stirring over a time span of 30 min at ambient temperature. Finally, the reaction mixture was purified by centrifugation[54]. The same procedure was used to prepare the LM-GO-II but, after centrifugation and resuspension in water, the concentration of GO in the dispersion was adjusted to 1.0 wt% after which the material was subjected to wet jet milling using a 0.1 mm nozzle at 150 MPa[55]. LM-GO-III was prepared in the same manner as the LM-GO-I but the graphite was oxidized twice under the same conditions. In the case of the LM-GO-IV, the synthesis procedure was the same as that used to prepare the LM-GO-II but the graphite was oxidized twice under the same conditions. Seven types of commercial GO were used: G-22L, G-21L, G-20-1, G-20L, G-17A, G-17S and G-19. In each procedure, 0.1 g $CoO_x$/BVO was dispersed in 150 mL of a 50.0 vol% aqueous solution of methanol (Wako) after which the GO dispersion was added. Unless noted otherwise, 0.8 wt% LM-GO-I with respect to the mass of $CoO_x$/BVO was added to the aqueous methanol solution, which contained 0.1 g $CoO_x$/BVO. The mixture was subsequently irradiated with the full spectrum output of a 300 W Xe lamp for 3 h after which the RGO/$CoO_x$/BVO was isolated by centrifugation.

### Photocatalytic $H_2$ or $O_2$ evolution

The photocatalytic hydrogen evolution trials were carried out in a Pyrex top-illuminated reaction vessel connected to a closed gas circulation system. Following the photodeposition of 0.5 wt% $Cr_2O_3$ with respect to the mass of STOS, a specific amount of $Cr_2O_3$/Pt/$IrO_2$/STOS (0.05, 0.1, 0.2, and 0.3 g) was redispersed in 150 mL of a 10.0 vol% aqueous methanol solution. The photoreaction conditions in trials using 0.2 g of STOS, $IrO_2$/STOS, $Cr_2O_3$/$IrO_2$/STOS, Pt/$IrO_2$/STOS, and $Cr_2O_3$/Pt/STOS were the same as those employed in trials with the $Cr_2O_3$/Pt/$IrO_2$/STOS. Photocatalytic oxygen evolution was performed by dispersing a specific amount of samples in 150 mL of a 50 mM

aqueous solution of $AgNO_3$ (Wako). After completely removing air from the system by evacuation, the background pressure was adjusted by introducing Ar gas to a preset pressure. Unless noted otherwise, the background pressure was set to 8.5 kPa for the $H_2$ and $O_2$ evolution half-reactions. Following these steps, each suspension was irradiated with a 300 W Xe lamp equipped with a cut-off filter (L42, $\lambda > 420$ nm) while the temperature of the reaction system was maintained at 288 K by circulating cooling water. The gaseous $H_2$ or $O_2$ evolved during each reaction was analyzed using an integrated thermal conductivity detector-gas chromatography system comprising a GC-8A gas chromatograph (Shimadzu Corp.) equipped with a 5 Å molecular sieve column, using Ar as the carrier gas.

Plots of $H_2$ generation via the hydrogen evolution reaction over time were fitted by assuming the rate of the forward reaction ($r_+$), meaning the hydrogen evolution reaction, was independent of the hydrogen partial pressure ($p$). This rate was therefore regarded as a constant ($I$) associated with the excitation intensity and AQY. The rate of the backward reaction (that is, the hydrogen oxidation reaction, ($r_-$), was also assumed to be proportional to the hydrogen partial pressure, with a proportionality coefficient (or rate constant) of $k$. Overall, the rate of change of the hydrogen partial pressure was calculated as $dp/dt = r_+ - r_- = I - kp$. This first-order linear non-homogeneous differential equation was solved as

$$p = -\frac{I}{k}\exp(-kt) + \frac{I}{k} \tag{1}$$

### Construction of Z-scheme system for OWS reaction
In a typical experiment, freshly prepared $Cr_2O_3/Pt/IrO_2/STOS$ (0.05 g) and $RGO/CoO_x/BVO$ (0.1 g) powders were dispersed in 150 ml of ultrapure water and irradiated by a 300 W Xe lamp equipped with a cut-off filter (L42, $\lambda > 420$ nm) for 12 h.

### Preparation of photocatalyst sheets
After the induction period and recovery by filtration, 0.02 g of dried $Cr_2O_3/Pt/IrO_2/STOS$-$RGO/CoO_x/BVO$ powder and 0.01 g of nanometer-sized silica particles (Nano Tek) were dispersed in 1 mL of ultrapure water and ultrasonicated for 5 min. Following this, the resulting suspension was drop cast onto a $3.0 \times 3.0$ cm glass substrate and then dried on a hot plate at 323 K.

### Photocatalytic Z-scheme OWS reactions using suspensions and sheets
After the induction period, the system was evacuated and the background pressure was adjusted to approximately 8.5 or 90 kPa by introducing Ar gas. The OWS reaction was subsequently performed using a setup similar to that employed during the $H_2$ evolution trials. The system was irradiated by a 300 W Xe lamp equipped with a cut-off filter (L42, $\lambda > 420$ nm). The performance characteristics of the Z-scheme system with $I_3^-/I^-$ or $Fe^{3+}/Fe^{2+}$ ionic couples were the same as that obtained using the solid-state electron mediator and with the HEP and OEP dispersed in 150 mL of 2.5 mM NaI or 2.0 mM $FeCl_3$ aqueous solutions, respectively. The pH was not adjusted in trials with the $I_3^-/I^-$ couple but was adjusted to 2.3 (using $H_2SO_4$) when working with the $Fe^{3+}/Fe^{2+}$ couple. Reactions were performed under simulated sunlight while covering the top window of the reactor with aluminum foil to ensure an illumination area of 9.0 cm$^2$ on the surface of the suspension system. The reactant solution was kept at 288 K by circulating cooling water during the reaction and the background pressure was adjusted to approximately 8.5 or 90 kPa after which the system was irradiated using an AM1.5 G solar simulator.

During experiments with photocatalyst sheets, the test method was the same as that detailed above but each sheet was placed at the bottom of a small reactor having an inner diameter of 4.5 cm and containing 50 mL of ultrapure water. For comparison purposes, 0.02 g of the Z-scheme powder was dispersed in 50 mL of water in the small reactor with an irradiation area of 9.0 cm$^2$ to set up a suspension system with similar reaction conditions to those applied when evaluating the sheet system.

### Apparent quantum yield measurements
AQY values were determined using a setup similar to that used to evaluate photocatalytic $H_2$ evolution but with irradiation by a 300 W Xe lamp equipped with various band-pass filters. The irradiation area was fixed at 9.0 cm$^2$ and the intensity of the monochromatic light was measured by a grating spectroradiometer. The AQY values for $H_2$ evolution, $O_2$ evolution, and OWS were calculated using the equation:

$$AQY(\%) = (AR)/I100 \tag{2}$$

where $R$ and $I$ denote the quantity of gas produced and the incident photon flux, respectively. In the case of $H_2$ evolution, the coefficient $A$ is the number of electrons consumed to generate a hydrogen molecule (i.e., 2) while for $O_2$ evolution, $A$ is the number of holes consumed to generate an oxygen molecule (i.e., 4). In the case of Z-scheme OWS, $A$ is 4 for $H_2$ evolution because photoexcitation occurs twice during the decomposition of water.

### Solar-to-hydrogen conversion efficiency measurements
The setup used to determine STH values was similar to that applied during the photocatalytic OWS trials but with irradiation by an AM1.5 G solar simulator. The STH conversion efficiency was obtained from the equation:

$$STH(\%) = (\boldsymbol{r_{H_2}} \times \Delta G_r)/(P \times S) \times 100 \tag{3}$$

where $\boldsymbol{r_{H_2}}$, $\Delta G_r$, $P$ and $S$ represent the hydrogen evolution rate during the Z-scheme OWS reaction, the Gibbs energy for the reaction $H_2O(l) \to H_2(g) + 1/2\, O_2(g)$, the energy intensity of the AM1.5 G solar irradiation (100 mW·cm$^{-2}$), and the irradiated area (9 cm$^2$), respectively. The value of $\Delta G_r$ used for the calculations was 237 kJ mol$^{-1}$ at 288 K[50].

### Characterization
Powder XRD analyses were performed with a Rigaku MiniFlex 300. UV-visible DRS data were acquired using a JASCO V-670 UV-visible system. XPS analyses were carried out with a PHI Quantera II instrument (ULVAC-PHI, Inc.) employing a monochromatized Al K$\alpha$ line source. The morphology of the samples was investigated using FE-SEM (Hitachi SU8000 and Phenom Pharos, ThermoFisher Scientific)[41]. TEM, SAED, HAADF-STEM, and EDS mapping were recorded using a JEOL JEM-2800 (JEOL) equipped with an X-MAX 100TLE SDD detector (Oxford Instruments) and a JEM-ARM200F Thermal Cs-STEM (JEOL). Cross-sectional cuts of the particles were prepared using an ion slicer (EM-09100IS, JEOL). Subsequently, a NanoMill Model 1040 (Fichione Instruments) was employed to remove the amorphous damage layers on TEM specimens prepared by the ion beam[56]. Elemental analysis of carbon was carried out on Horiba's EMIA-Pro. Mott-Schottky plot was performed in 0.1 mol·L$^{-1}$ $Na_2SO_4$ aqueous solution on BioLogic VSP-300 electrochemical system and the STOS electrode was prepared by the particle transfer method with Ti substrate[13].

### Transient absorption spectroscopy
The dynamical behavior of photogenerated charge carriers in the STOS and BVO photocatalysts was studied using a custom-built pump-probe system equipped with Nd:YAG lasers (Continuum, Surelite I) and custom-built spectrometers. TA spectra were acquired from 6000 cm$^{-1}$ (approximately 0.74 eV) to 1200 cm$^{-1}$ (approximately 0.15 eV). Photo-carrier generation was induced by irradiating the sample with 470 nm laser pulses (duration: 6 ns, fluence: 0.1 or 1 mJ/pulse, frequency: 1 Hz) with one laser shot applied every second. Photogenerated electrons

were probed in the mid-IR region by focusing the probe beam originating from the $MoSi_2$ coil on the sample after which the transmitted infrared beam was introduced into a grating spectrometer. The monochromated light was detected using a mercury cadmium telluride detector (Kolmar). The decay profile associated with transient absorption at $5000\ cm^{-1}$ was monitored to further investigate the decay behavior of photoinduced electrons in both bare and cocatalyst-loaded STOS. Photoinduced holes in the BVO and $CoO_x$-loaded BVO were monitored at 505 nm. In these trials, a probe beam from a halogen lamp was focused on the sample, and the reflected light passed through a grating spectrometer and was then detected by Si photodiodes. The output electrical signal was subsequently amplified with an AC-coupled amplifier (Stanford Research Systems, SR560, 1 MHz) capable of tracking responses on time scales from 1 microsecond to several milliseconds. A total of 600 responses was averaged to obtain the transient decay profile at the probe wavelength. The time resolution of the spectrometer was limited to 1 μs by the bandwidth of the amplifier. Specimens for TAS assessments were prepared by dispersing the photocatalyst powder in water and then drop-casting the suspension on a $CaF_2$ substrate. This substrate was then dried in air overnight to obtain a powder film with a density on the order of $1.09\ mg/cm^2$. The as-prepared $CaF_2$/film specimen was placed in a stainless steel reaction cell that was subsequently evacuated before exposing the film to $N_2$ gas at a pressure of 20 Torr prior to the measurement. The measurements were carried out at room temperature.

**Water reactivity measurements of the Z-scheme system by TAS with steady-state irradiation**

In this measurement, a modified FTIR spectrometer equipped was used with an irradiation system for steady-state pumping. The aperture size of the IR beam (for probing) was set at 4 mm. The transmitted IR beam from the sample was detected by the MCT detector. The accumulated electrons in the sample were probed at 0.1 eV. For sample preparation, it is similar to the procedure employed in the TAS measurement described above. The as-prepared film/$CaF_2$ was placed in a stainless-steel reaction cell and subsequently evacuated before exposing $N_2$ gas or $H_2O$ vapor (cell pressure: 10 Torr) prior to the measurement. The samples were irradiated by 470 nm continuous wave (CW)-LED and the measurement were carried out at room temperature.

## Data availability

The data that support the findings of this study are available from the corresponding author upon reasonable request. Source data are provided with this paper.

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

## Acknowledgements

This research was supported by the Artificial Photosynthesis Project (ARPChem) of the New Energy and Industrial Technology Development Organization (NEDO). This work was partially supported by "Advanced Research Infrastructure for Materials and Nanotechnology in Japan (ARIM)" of the Ministry of Education, Culture, Sports, Science and Technology (MEXT), Grant Number JPMXP1222UT0023. The authors thank Ms. Michiko Obata of Shinshu University for her assistance during the XPS analyses.

## Author contributions

L.L., Y.M., T.H., T.T., and K.D. conceived and designed the experiments. Y.M., C.G., and X.T. synthesized the experimental samples. Y.N. produced the GO solution for the experiments. L.L. and Y.M. carried out the majority of the characterizations and photocatalytic reactions. H.Y. developed the method for surface modification of the HEP. L.L. and Y.M. performed the SEM observations. M.N. and N.S. carried out the TEM characterizations. J.J. M.V., and A.Y. performed the TAS analyses. Y.P. carried out the photoelectrochemical measurements. K.D. supervised the research. All authors discussed the results and participated in writing the manuscript.

## Competing interests

The authors declare no competing interests.
