## [Peer Review File · Nature Communications]

Efficient and stable visible-light-driven Z-scheme overall water splitting using an oxysulfide H₂ evolution photocatalystREVIEWER COMMENTS

Reviewer #1 (Remarks to the Author):

In this manuscript, Domen and co-workers applied Sm₂Ti₂O₅S₂ (STOS) and BiVO₄ (BVO) to form Z-Scheme photocatalytic system for overall water splitting. H₂ evolution reaction was achieved by STOS and O₂ evolution reaction was performed by BVO. Author started with half-reactions, for instance, a AQY of STOS for hydrogen evolution reaction is ca. 21.7% (420 nm), and optimal AQY of BVO for oxygen evolution reaction is 29.5% (420 nm). Overall water splitting was then realized under visible light illumination, and impressively, this system can be achieved with the pressure of to 90 kPa. This result is of great important for practical use of overall water splitting by sunlight illumination. This manuscript is well written and merits the general readers of Nat. Commun. I would suggest accepting this manuscript by minor corrections.

(1) STOS in this system is carefully optimized to achieve this result, and how about the activity of STOS in H₂ evolution half-reaction if only CaCl₂ or LiCl is used as the flux?

(2) It is interesting that STOS prepared by CsCl flux showed no activity in H₂ evolution half-reaction. Can the authors provide more characterization data of STOS-CsCl to reveal possible defects inside the crystal, such as XPS and TEM?

(3) Considering that not all of the GO were loaded on BVO, can you define the loading amount of GO on BVO?

(4) What happen if increased the amount of photodeposited Cr₂O₃? Does the reverse reaction been inhibited?

(5) The reduction of the photoactivity is owing to the adhesion of the photocatalyst powder to the reactor wall, can author determine the amount of the photocatalyst on the reactor wall?

Reviewer #2 (Remarks to the Author):

The authors developed a high-performance oxysulfide photocatalyst (Sm₂Ti₂O₅S₂) as a hydrogen evolution photocatalyst, and combined it with BiVO₄ and reduced graphene oxide to achieve a STH of 0.22% of overall water splitting. Unlike many other systems, this setup can proceed water splitting efficiently under increased background pressure, indicating its suitability for practical applications. The paper could be published after the authors address the following issues:

1) In figure 1e and 1f, why is only Pt clustered in a round shape, unlike other elements? The Pt can plays a key role in H₂ evolution reaction. Is there a possibility of increasing efficiency if it is deposited flat?

2) In figure 3a, why does the efficiency decrease when the amount of Cr₂O₃/Pt/IrO₂/STOS is more than 0.2g?

3) In figure 4b, why does the efficiency drastically decrease under high background pressure?

4) Is there any optimizing experiment results for the amounts of Pt, IrO₂, and Cr₂O₃ deposition?

Reviewer #3 (Remarks to the Author):

In this work, the author developed a high-performance oxysulfide photocatalyst, Sm₂Ti₂O₅S₂, as a HEP for use in a Z-scheme OWS system in combination with BiVO₄ as the OEP and rGO as the solid-state electron mediator. This system was able to split water into hydrogen and oxygen for more than 100 hours with a STH energy conversion efficiency of 0.22%. However, the novelty and scientific details are not high enough compared to their previously reported works. The detailed comments are as follows:

1. The authors claimed that rGO acts as a solid-state electron mediator to accelerate the carrier migration and thus enhance the performance of photocatalytic water splitting. However, this claim was not confirmed in this manuscript. It is suggested that the authors incorporate a comparative analysis of carrier kinetics between STOS/BVO and STOS/rGO/BVO to clarify the role of rGO.
2. The author used many co-catalysts (Pt, Cr₂O₃, IrO₂, CoOx) in the catalyst preparation process, among which Pt is commonly used as a co-catalyst for hydrogen production, Cr₂O₃ is used to suppress the reverse reaction, and CoOx is used as a co-catalyst for oxygen production. What is the role of IrO₂ loaded on STOS? Moreover, loading IrO₂ on STOS did not significantly improve its hydrogen production performance, but the overall water splitting performance of the heterojunction constructed using IrO₂-STOS as HEP was nearly doubled. The reason behind this phenomenon needs elucidation.
3. The authors should supplement the changes of Ti, Bi and other elements before and after the formation of the Z-scheme heterojunction. The formation of heterojunction may change the binding energy of elements in STOS and BVO.
4. Charge transfer mechanism in Z-scheme heterojunction is influenced by the band structure of HEP and OEP. However, the authors omitted an analysis of the energy band structure for STOS, BVO, and the resultant Z-scheme heterojunction. To ensure reasonability of Z scheme mechanism, an exploration of these energy band structures is essential.
5. The author provided the SEM image of Cr₂O₃/Pt/IrO₂/STOS-RGO/CoOx/BVO catalyst, but it is difficult to distinguish STO and BVO in this image. It is recommended that the author supplement the corresponding EDX mapping.
6. In this article, the author prepared this Cr₂O₃/Pt/IrO₂/STOS-RGO/CoOx/BVO Z-scheme photocatalyst with a very complicated process and expensive materials, but the achieved performance is lower than that of other materials (STH is only 0.22%). This photocatalyst appears to lack major innovations in design and execution.

Response to Reviewers

Reviewer #1

In this manuscript, Domen and co-workers applied $\text{Sm}_2\text{Ti}_2\text{O}_5\text{S}_2$ (STOS) and BiVO_4 (BVO) to form Z-Scheme photocatalytic system for overall water splitting. H_2 evolution reaction was achieved by STOS and O_2 evolution reaction was performed by BVO. Author started with half-reactions, for instance, a AQY of STOS for hydrogen evolution reaction is ca. 21.7% (420 nm), and optimal AQY of BVO for oxygen evolution reaction is 29.5% (420 nm). Overall water splitting was then realized under visible light illumination, and impressively, this system can be achieved with the pressure of to 90 kPa. This result is of great important for practical use of overall water splitting by sunlight illumination. This manuscript is well written and merits the general readers of Nat. Commun. I would suggest accepting this manuscript by minor corrections.

--- We appreciate Reviewer 1's support toward the publication of our work.

(1) STOS in this system is carefully optimized to achieve this result, and how about the activity of STOS in H_2 evolution half-reaction if only CaCl_2 or LiCl is used as the flux?

Response: Thank you for your question. We prepared STOS by using CaCl_2 or LiCl flux under the same synthesis conditions, which were referred to as STOS- CaCl_2 and STOS- LiCl , respectively. According to the XRD analysis, STOS could be obtained as the major product, the impurities such as TiO_2 and $\text{Sm}_2\text{Ti}_2\text{O}_7$ were found in the STOS- CaCl_2 and STOS- LiCl samples (**Figure R1a**). When performing the H_2 evolution half-reaction under the same conditions, the STOS- CaCl_2 and STOS- LiCl showed lower activity than STOS prepared using a $\text{CaCl}_2/\text{LiCl}$ eutectic mixture, referred to as STOS, as shown in **Figure R1b**. We further analyzed the chemical state and morphology of STOS- CaCl_2 and STOS- LiCl . No obvious difference was found by the XPS analysis (**Figure R2**), indicating that the surface state of STOS was unchanged regardless of the kinds of the flux reagents. However, the sheet-like STOS- CaCl_2 particles were larger and more irregular, forming agglomerates, compared with STOS prepared with the $\text{CaCl}_2/\text{LiCl}$ mixture (**Figure R3b**). In addition, impurity TiO_2 particles were also observed. Similar results were observed in the case of STOS- LiCl (**Figure R3c**). The higher melting points of CaCl_2 (772 °C) and LiCl (605 °C) than that of the $\text{CaCl}_2/\text{LiCl}$ eutectic mixture (475 °C) may account for the low crystallinity and photocatalytic activity. We have included the data in the revised manuscript (Line 22, Page 6; Line 22, Page 7) and supporting information (Supplementary Fig. 7-9).

Figure R1. (a) XRD patterns of STOS-LiCl and STOS-CaCl₂. (b) Amounts of H₂ evolved over STOS, STOS-LiCl and STOS-CaCl₂ specimens with surface modifications of Cr₂O₃/Pt/IrO₂ over time.

Figure R2. XPS pattern of (a) Sm 3d, (b) Ti 2p, (c) O 1s and (d) S 2s of STOS, STOS-LiCl and STOS-CaCl₂.

Figure R3. SEM images of (a) STOS, (b) STOS-CaCl₂ and (c) STOS-LiCl.

(2) It is interesting that STOS prepared by CsCl flux showed no activity in H₂ evolution half-reaction. Can the authors provide more characterization data of STOS-CsCl to reveal possible defects inside the crystal, such as XPS and TEM?

Response: Thank you for your suggestion. We carried out XPS and TEM analysis on the sample of STOS prepared with CsCl flux. As show in **Figure R4**, the XPS peak of Sm $3d$ and Ti $2p$ were slightly shifted to higher and lower binding energy, respectively, compared with those in the STOS sample prepared with the $\text{CaCl}_2/\text{LiCl}$ flux. In addition, the peak intensity of adsorbed oxygen species was greatly increased. Two additional S $2s$ peaks located at 231.4 and 227.5 eV were observed. The former can be ascribed to the formation of SO_4^{2-} , while the latter was assigned to the adsorbed H_2S as a strong odor of H_2S was noticed during the acid post-treatment. In contrast to STOS- CaCl_2 and STOS- LiCl , the surface state of STOS- CsCl was significantly different from STOS prepared with the $\text{CaCl}_2/\text{LiCl}$ flux. Additionally, **Figure R5a and b** shows dark-field TEM images of STOS- CsCl , where numerous dislocations were observed on the (001) surface. High resolution TEM image further confirmed the presence of defects inside the crystal (**Figure R5c and d**). These results indicate that STOS prepared with the CsCl flux had poor crystallinity. We have included the data in the revised manuscript (Line 10, Page 6) and supporting information (Supplementary Fig. 5 and 6). Thank you.

Figure R4. XPS pattern of (a) Sm $3d$, (b) Ti $2p$, (c) O $1s$ and (d) S $2s$ of STOS and STOS- CsCl .

Figure R5. (a, b) Dark field TEM images of STOS-CsCl. (c, d) Atomic resolution ADF images STOS-CsCl.

(3) Considering that not all of the GO were loaded on BVO, can you define the loading amount of GO on BVO?

Response: Thank you for your question. To determine the actual loading amount of RGO, we performed the elemental analysis using a carbon/sulfur bulk content analyzer. The amount of carbon loaded was greater than the nominal amount of the GO added. Therefore, we checked the concentration of the GO solution used. It was found that the actual concentration of the GO stock solution was 1.6 times higher than the nominal concentration, probably due to the variation in sample preparation, inhomogeneity of the stock solution, or evaporation of the solvent (water) during storage. Accordingly, the added amount of GO was revised throughout the manuscript. As shown in Table R1, the loading amount of carbon was gradually increased with the increased amount of added GO. The remaining GO will be removed during the centrifugation procedure because not all the GO was deposited at higher added amount. We have included the data in the revised manuscript (Line 21, Page 10) and supporting information (Supplementary Table 1). Thank you.

Table R1. The amounts of added GO and carbon loaded on the Z-scheme system.

Added amount (wt% vs. BVO)	Detected amount in the Z-scheme system (wt% vs. total mass)	Detected amount in Z-scheme system (wt% vs. BVO)
0.3	0.3	0.4
0.8	0.4	0.6
1.6	0.8	1.1
3.2	1.7	2.5

(4) What happen if increased the amount of photodeposited Cr₂O₃? Does the reverse reaction been inhibited?

Response: Thank you for your question. We investigated the impact of loading amount of Cr₂O₃ on the performance. The reverse reaction was in fact inhibited by increasing the amount of photodeposited Cr₂O₃ beyond 0.5 wt% (**Figure R6a**). However, at the same time, the H₂ evolution performance was gradually decreased with increasing the Cr₂O₃ loading amount (**Figure R6b**). Concurrently, the performance of overall water splitting was also decreased (**Figure R6c**). Therefore, photodeposition of 0.5 wt% Cr was an optimal amount to balance forward and reverse reaction. We have included the data in the revised manuscript (Line 3, Page 19) and supporting information (Supplementary Fig. 32). Thank you.

Figure R6. (a) Back reaction of the Z-scheme system using Pt/IrO₂/STOS loaded with different amounts of Cr₂O₃ under dark condition. (b) H₂ evolution rate of Pt/IrO₂/STOS loaded with different amounts of Cr₂O₃. (c) H₂ and O₂ evolution rates of the Z-scheme system using Pt/IrO₂/STOS loaded with different amounts of Cr₂O₃.

(5) The reduction of the photoactivity is owing to the adhesion of the photocatalyst powder to the reactor wall, can author determine the amount of the photocatalyst on the reactor wall?

Response: Thank you for your question. After photoreaction for one run (12 h), the adhesion of the photocatalyst powder to the reactor wall was visually confirmed (**Figure R7**). We carefully collected the photocatalyst powder in the solution by filtration, followed by drying in the vacuum oven. The procedure was repeated three times and the average amount of remaining photocatalyst was 146.8 mg. This means that in one run of the photoreaction for 12 h, 3.2 mg of the photocatalyst would adhere to the reactor wall. After each run, a similar amount of the photocatalyst powder would adhere to the reactor wall. During 8 runs for a total of 114 h, the amount of powder lost would be around 24 mg. This will reasonably account for the decrease in the activity. We have included the data in the revised manuscript (Line 6, Page 16) and supporting information (Supplementary Fig. 28).

Figure R7. Photograph of the reaction after one run of 12 h.

Reviewer #2

The authors developed a high-performance oxysulfide photocatalyst ($\text{Sm}_2\text{Ti}_2\text{O}_5\text{S}_2$) as a hydrogen evolution photocatalyst, and combined it with BiVO_4 and reduced graphene oxide to achieve a STH of 0.22% of overall water splitting. Unlike many other systems, this setup can proceed water splitting efficiently under increased background pressure, indicating its suitability for practical applications. The paper could be published after the authors address the following issues:

--- We appreciate Reviewer 2's support toward the publication of our work.

1) In figure 1e and 1f, why is only Pt clustered in a round shape, unlike other elements? The Pt can play a key role in H_2 evolution reaction. Is there a possibility of increasing efficiency if it is deposited flat?

Response: Thank you for your question. In addition to the microwave deposition method, we loaded Pt by impregnation and photodeposition methods and investigated the morphology. As shown in **Figure R8a-c**, the Pt cluster with round shape was formed in all cases. Therefore, it seems that after the initial chemical or photocatalytic reduction of Pt^{4+} ions to metallic Pt on the surface, the subsequent Pt^{4+} ions tend to be reduced on the Pt metal rather than on the surface of STOS to form round clusters (**Figure R8d**). At present, we do not have feasible methods to deposit flat Pt cocatalyst particles, and we will take into account the dependence of performance on the cocatalyst morphology in our future work. Thank you for your understanding.

Figure R8. SEM images of Pt-loaded STOS using (a) microwave-assisted heating, (b) impregnation-hydrogen reduction and (c) photodeposition and (d) a schematic diagram of the formation of round Pt clusters.

2) In figure 3a, why does the efficiency decrease when the amount of $\text{Cr}_2\text{O}_3/\text{Pt}/\text{IrO}_2/\text{STOS}$ is more than 0.2g?

Response: Thank you for your question. Two major reasons can account for the decrease H₂ evolution performance when the amount of Cr₂O₃/Pt/IrO₂/STOS is more than 0.2 g. Firstly, the high concentration of the photocatalyst powder will block the incident light due to the scattering at the top part of the suspension, which would lower the number of photons absorbed by the photocatalyst. Second, an increase in the amount of Cr₂O₃/Pt/IrO₂/STOS also increases the amount of Pt cocatalyst that accelerates the hydrogen oxidation reaction (**Supplementary Figure 30c and d**), which competes with the H₂ evolution forward reaction. We have included the discussion in the revised manuscript (Line 11, Page 8; Line 11, Page 18). Thank you.

3) In figure 4b, why does the efficiency drastically decrease under high background pressure?

Response: Thank you for your question. There is a misunderstanding of the data. In fact, the efficiency did not decrease appreciably under high background pressure. In Figure 4b, the overall water splitting reaction was carried out for 18 h in the first run with a background pressure of 8.5 kPa. In the second run, the reaction was continued for only 12 h at a background pressure of 90 kPa. The H₂ and O₂ evolution rates are comparable in the same reaction time.

4) Is there any optimizing experiment results for the amounts of Pt, IrO₂, and Cr₂O₃ deposition?

Response: Thank you for your question. We have included the results for optimizing the loading amounts of Pt, Cr₂O₃, and IrO₂ (Line 6, Page 8, and Supplementary Fig. 12). It is well known that Pt serves as a H₂ evolution active site during the water splitting reaction. Therefore, the loading amount of Pt was optimized firstly (**Figure R9a**). Then, the deposition amount of Cr₂O₃ was optimized. The loading of Cr₂O₃ significantly improved the H₂ evolution activity (**Figure R9b**). Finally, the amount of IrO₂ was varied from 0.3 wt% to 1.0 wt% and the highest performance was obtained by loading 0.5 wt% of IrO₂ (**Figure R9c**).

Figure R9. H₂ evolution rate of STOS (a) loaded with different amounts of Pt by the microwave deposition method, (b) loaded with different amounts of Cr by the photodeposition method after loading 1.0 wt% Pt, and (c) STOS loaded with different amounts of Ir by the microwave deposition method and subsequently with 1.0 wt% Pt and 0.5 wt% Cr.

Reviewer #3 (Remarks to the Author):

In this work, the author developed a high-performance oxysulfide photocatalyst, $\text{Sm}_2\text{Ti}_2\text{O}_5\text{S}_2$, as a HEP for use in a Z-scheme OWS system in combination with BiVO_4 as the OEP and rGO as the solid-state electron mediator. This system was able to split water into hydrogen and oxygen for more than 100 hours with a STH energy conversion efficiency of 0.22%. However, the novelty and scientific details are not high enough compared to their previously reported works. The detailed comments are as follows:

--- We appreciate Reviewer 3's helpful comments and suggestions. This study contains three significant advances. First, the H_2 evolution performance of STOS has been greatly enhanced with a factor of 36 times compared with the previous works. This is in fact one of the highest ever achieved using visible-light active non-oxide photocatalysts. Second, the as-constructed Z-scheme system shows a dramatically improved STH value, which is 73 times higher than previous reported STOS-based Z-scheme system and also one of the best among the systems of $\text{SrTiO}_3\text{-WO}_3$, doped $\text{SrTiO}_3\text{-BiVO}_4$, TaON-BiVO_4 and $\text{BaTaO}_2\text{N-WO}_3$. Third, this Z-scheme system maintains the performance under near atmospheric pressure, which is a critical characteristic toward the practical operation of photocatalytic water splitting. This work also demonstrates that it is possible to dramatically improve the performance of the Z-scheme system by independently improving the activity of HEP and also the importance of charge transfer between HEP and OEP is confirmed by TAS analysis. Moreover, the back reaction is studied in detail, which will guide the future development of Z-scheme system. We believe those progresses guarantee the novelty and impact of present work. In addition, we have revised the manuscript according to your and other reviewers' valuable comments to provide more scientific details and improve the quality of our manuscript. Thank you.

1. The authors claimed that rGO acts as a solid-state electron mediator to accelerate the carrier migration and thus enhance the performance of photocatalytic water splitting. However, this claim was not confirmed in this manuscript. It is suggested that the authors incorporate a comparative analysis of carrier kinetics between STOS/BVO and STOS/rGO/BVO to clarify the role of rGO.

Response: Thank you for your suggestion. To support the enhancement of charge transfer by RGO in the Z-scheme system, we carried out the measurement of TAS after steady-state irradiation in the absence and presence of water vapor in a nitrogen atmosphere for the Z-scheme system with and without RGO (**Figure R10**). Note that the accumulated electrons can participate in the water reduction process in the presence of water vapor. In the absence of RGO, the decay of electrons was hardly accelerated by the introduction of water vapor, indicating a low efficiency to the water reduction reaction by photoexcited electrons. Interestingly, when RGO was incorporated in the Z-scheme system, the decay of electrons was much accelerated by the exposure to water vapor. This means that the reactivity of the accumulated electrons for water reduction was significantly enhanced by the RGO promoting the formation of the Z-scheme. These results demonstrated the connection of HEP and OEP with RGO can significantly promote the charge transfer between HEP and OEP. We have

included the data in the revised manuscript (Line 10, Page 13) and supporting information (Supplementary Fig. 23). Thank you.

Figure R10. Normalized decay kinetics of accumulated electrons in the Z-scheme system without (a) and with RGO (b) measured in nitrogen (10 Torr) and water vapor (10 Torr). The samples were excited under continuous irradiation for 223 s using 470 nm CW-LED (Fluence: 40 mW/cm^2) before the measurement of the signal decay.

2. The author used many co-catalysts (Pt, Cr_2O_3 , IrO_2 , CoOx) in the catalyst preparation process, among which Pt is commonly used as a co-catalyst for hydrogen production, Cr_2O_3 is used to suppress the reverse reaction, and CoOx is used as a co-catalyst for oxygen production. What is the role of IrO_2 loaded on STOS? Moreover, loading IrO_2 on STOS did not significantly improve its hydrogen production performance, but the overall water splitting performance of the heterojunction constructed using IrO_2 -STOS as HEP was nearly doubled. The reason behind this phenomenon needs elucidation.

Response: Thank you for your question. To explore the role of IrO_2 , we measured the O_2 evolution activity of STOS modified with IrO_2 , Pt/IrO_2 , $\text{Cr}_2\text{O}_3/\text{Pt/IrO}_2$ and $\text{Cr}_2\text{O}_3/\text{Pt}$. As shown in **Figure R11**, the O_2 evolution performance was significantly improved when the IrO_2 was loaded. The enhancement of activity in both the O_2 and H_2 evolution half-reactions indicates that IrO_2 can capture photogenerated holes and therefore promote charge separation, resulting in the enhanced performance of HEP in overall water splitting. However, in H_2 evolution half reaction, IrO_2 is not an efficient cocatalyst for oxidizing the sacrificial reagent. Therefore, the H_2 evolution rate was not significantly enhanced after co-loading of IrO_2 . We have included the data in the revised manuscript (Line 2, Page 15) and supporting information (Supplementary Fig. 25b). Thank you.

Figure R11. Oxygen evolution activity of STOS loaded with different cocatalysts.

3. The authors should supplement the changes of Ti, Bi and other elements before and after the formation of the Z-scheme heterojunction. The formation of heterojunction may change the binding energy of elements in STOS and BVO.

Response: Thank you for your suggestion. Because the STOS and BVO were not formed intimate heterojunction and their crystals were loosely connected by RGO sheets, it is unlikely that the binding energies of Ti and Bi in STOS and BVO, respectively, are changed by the formation of the Z-scheme system. In fact, according to the XPS analysis, the peak positions of Ti *2p* orbitals were nearly the same before and after the formation of Z-scheme system (**Figure R12a; Supplementary Figure 2**). On the other hand, the peak position of Bi *4f* appear to have shifted to higher binding energy after photodeposition of RGO, but not further by the formation of Z-scheme system (**Figure R12b; Supplementary Figure 10**).

Figure R12. XPS patterns of (a) Ti *2p* and (b) Bi *4f* of the samples before and after the formation of Z-scheme system.

4. Charge transfer mechanism in Z-scheme heterojunction is influenced by the band structure of HEP and OEP. However, the authors omitted an analysis of the energy band structure for STOS, BVO, and the resultant Z-scheme heterojunction. To ensure reasonability of Z scheme

mechanism, an exploration of these energy band structures is essential.

Response: Thank you for your suggestion. To determine the conduction band minimum (CBM) of STOS, we have performed the Mott-Schottky measurement. As shown in **Figure R13a**, the CBM of STOS was measured to be around -0.6 V vs. NHE. According to the bandgap of 1.9 eV determined by UV-vis DRS, the valance band maximum (VBM) located at around 1.3 V vs. NHE. In combination with the reported band position of well-developed BiVO₄ (Joule 2018, 2, 2393-2402; Nat. Commun. 2022, 13, 484), the energy band structures at pH 6.8 are presented in **Figure R13b**. The band alignment allows the charge transfer in the Z-scheme manner. We have included the data in the revised manuscript (Line 12, Page 9) and supporting information (Supplementary Fig. 13). Thank you.

Figure R13. (a) Mott-Schottky plot of STOS. (b) Band alignment of the Z-scheme system.

5. The author provided the SEM image of Cr₂O₃/Pt/IrO₂/STOS-RGO/CoOx/BVO catalyst, but it is difficult to distinguish STO and BVO in this image. It is recommended that the author supplement the corresponding EDX mapping.

Response: Thank you for your suggestion. We performed EDX mapping of the Z-scheme system. As shown in **Figure R14**, the STOS and BVO can be identified by the distribution of Sm and Bi elements, respectively. Besides, in order to clearly observe RGO sheets, a secondary electron detector was used for imaging. However, when a back scattering electron detector was used for imaging, it is possible to distinguish particle of STOS and BVO by the grayscale contrast. This is because the signal for back scattered electrons becomes stronger with the increased atomic number of the element in the compound. The atomic number of Bi is greater than Sm and therefore BVO particles appear brighter than STOS particles (**Figure R15**). We have included the data in the revised manuscript (Line 5, Page 10) and supporting information (Supplementary Fig. 16 and 17). Thank you.

Figure R14. SEM image of the Z-scheme system and corresponding EDS mapping.

Figure R15. SEM images of Z-scheme system with (a) back scattering detector and (b) secondary electron detector modes.

6. In this article, the author prepared this $\text{Cr}_2\text{O}_3/\text{Pt}/\text{IrO}_2/\text{STOS-RGO}/\text{CoO}_x/\text{BVO}$ Z-scheme photocatalyst with a very complicated process and expensive materials, but the achieved performance is lower than that of other materials (STH is only 0.22%). This photocatalyst appears to lack major innovations in design and execution.

Response: Thank you for your insightful comment on the future development of as-constructed Z-scheme system. We reply your concerns from the following three aspects:

(a) In fact, the Z-scheme system is more complicated than the one-step excitation overall water splitting system. However, most of the photocatalysts that can achieve one-step excitation overall water splitting respond only to UV light, which negates their practical application in the future due to insufficient theoretical STH. Only few visible-light-responsive photocatalysts can realize one-step excitation overall water splitting, and most of them show low performance at present. The Z-scheme system is an alternative and promising pathway, because it allows the combination of visible-light-responsive photocatalysts that are highly active in the hydrogen evolution half-reaction and the oxygen evolution half-reaction to realize overall water splitting. The HEP, OEP and electron mediator can be optimized independently to improve the overall efficiency of the Z-scheme system, as demonstrated in this work. In addition, the spatial separation of H₂ and O₂ evolution sites in the Z-scheme system can expand the possibilities for charge separation and suppression of reverse reaction. These advantages of Z-scheme system can offset its complexity and rather represents much room for research and development for further understanding and improvement. Our work will help establish guidelines for the seemingly complex Z-scheme system and guide research in the right direction.

(b) Although higher STH values have been reported previously, most of these have been recorded at reduced pressure, and high performance has rarely been observed at ambient pressure, which is assumed to be a practical operating condition. The nearly pressure-independent nature of the present system makes the Z-scheme system more favorable for the test under practical operating conditions. This work also specified the causes of backward reactions and highlights the importance of solving the problem in the future study. In addition, the STH of the current system is mainly limited by the OEP due to its relatively wide bandgap. If a suitable OEP with a long wavelength edge wavelength is developed, the performance can be further improved.

(c) Finally, the amount of photocatalysts per square meter in the construction of water splitting sheet is quite small. Thus, in practice, the reactors will cost more than the photocatalyst materials, and the use of noble metals as cocatalysts will not be the main issue. In addition, the photocatalysts and noble metal co-catalysts are not consumed during photocatalytic water splitting as long as they are stable, and they can be recycled after use. The noble metal cocatalysts, such as Pt and Ir, can potentially be replaced with other non-noble metal cocatalysts under development. All these factors will lower the cost of using the Z-scheme system in large-scale applications.

The aforementioned three significant advances have been included in the manuscript, which guarantee the innovation of this work. Moreover, this work demonstrates that it is possible to dramatically promote the performance of the Z-scheme system by optimizing the HEP and the electron transfer between HEP and OEP. Therefore, there is still much room for

improvement in the performance toward practical application by exploring suitable OEP and inhibiting the reverse reaction. We have included additional discussion in the revised manuscript (Line 14, Page 3; Line 22, Page 4; Line 17 and 21, Page 20) to highlight the innovations and feasibility of the present system.

REVIEWERS' COMMENTS

Reviewer #1 (Remarks to the Author):

The revised manuscript is recommended for acceptance.

Reviewer #2 (Remarks to the Author):

The authors have made satisfactory revisions. The manuscript should be published without further delay.

Reviewer #3 (Remarks to the Author):

With all the comments well addressed, the revised version is acceptable.

Response to referees

All three reviewers have recommended that this manuscript be published as is. We would like to thank them for their time and effort.

Reviewer #1 (Remarks to the Author):

The revised manuscript is recommended for acceptance.

Reviewer #2 (Remarks to the Author):

The authors have made satisfactory revisions. The manuscript should be published without further delay.

Reviewer #3 (Remarks to the Author):

With all the comments well addressed, the revised version is acceptable.